# Cobalt-catalyzed difluoroalkylation of tertiary aryl ketones for facile synthesis of quaternary alkyl difluorides

Chao Li [1], Yi-Xuan Cao[1], Rui Wang[1], Yi-Ning Wang[1], Quan Lan[1] & Xi-Sheng Wang [1]

The selective incorporation of *gem*-difluoroalkyl groups into biologically active molecules has long been used as an efficient strategy for drug design and discovery. However, the catalytic $C(sp^3)$-$CF_2$ bond-forming cross-coupling reaction for selective incorporation of difluoromethylene group into diverse alkyl chains, especially more sterically demanding secondary and tertiary functionalized alkanes, still remains as a major challenge. Herein, we describe a cobalt-catalyzed difluoroalkylation of tertiary aryl ketones for facile synthesis of quaternary alkyl difluorides, which exhibited high efficiency, broad scope and mild conditions. The synthetic utility of this method is demonstrated by late-stage difluoroalkylation of donepezil, a well-known acetylcholinesterase inhibitor used to treat the Alzheimer's disease. Preliminary mechanistic investigations indicate that a difluoroalkyl radical is involved in a Co(I)/Co(III) catalytic cycle. This cobalt-catalyzed fluoroalkylation thus offers insights into an efficient way for the synthesis of fluoroalkylated bioactive molecules for drug discovery.

[1] Hefei National Laboratory for Physical Sciences at the Microscale and Department of Chemistry, Center for Excellence in Molecular Synthesis of CAS, University of Science and Technology of China, Hefei, Anhui 230026, China. These authors contributed equally: Chao Li, Yi-Xuan Cao. Correspondence and requests for materials should be addressed to X.-S.W. (email: xswang77@ustc.edu.cn)

The wide use of fluorinated compounds in pharmaceuticals, agrochemicals, and advanced materials vigorously propelled the development of efficient methods for the incorporation of fluorine or fluorinated moieties into organic molecules[1–3]. In the past decades, transition-metal-catalyzed fluoroalkylation has emerged as an efficient and convenient alternative approach to direct fluorination for facile synthesis of fluorine-containing organic compounds[4–11]. Among all well-developed active catalysts, palladium-[12-19] and copper-catalyzed[20–29] or -mediated[30–33] fluoroalkylation reactions have long proved to be the most useful and reliable tools for effective synthesis of fluoroalkylated molecules. Recently, the earth-abundant, inexpensive, and environmentally friendly first-row transition metals, including Ni, Fe, and Co, have attracted increasing attention on C–C bond-forming reactions due to their different and complementary catalytic reactivities to late and noble transition metals[34–36]. Whereas nickel[37–44] and iron[45–48] have already been demonstrated as efficient catalysts for such fluoroalkylation reactions, cobalt-catalyzed fluoroalkylation has been less studied and remained as a major challenge. To the best of our knowledge, the only two examples on cobalt-catalyzed[49,50] fluoroalkylation were limited to aryl zinc and magnesium nucleophiles, whereas the cross-coupling of alkyl species for construction of fluorinated alkanes still remained as an issue left to be solved so far.

The selective incorporation of *gem*-difluoroalkyl groups into drug-like molecules has long been known as a powerful strategy for drug design and discovery, as their structural diversity and the ability to modulate the electronic properties of parent molecules[51–54]. To date, a number of difluoroalkylation reactions have been well-developed into the construction of C(sp$^2$ or sp)-CF$_2$R bonds, in which fluorine atoms were normally oriented at relatively unique benzyl, allyl, and propargyl positions[12–20,23–28,46–48,55–57]. In contrast, the catalytic C(sp$^3$)-CF$_2$ bond-forming reaction for selective incorporation of difluoromethylene group into the alkyl chain at any special position, is still scarce and remains as an unsolved problem[58]. The only example on transition-metal catalyzed cross-coupling between alkyl substrates and difluoroalkylating reagents was just reported by Zhang and coworkers[59]; however, the alkylzinc reagents in this transformation were still limited to primary alkyl compounds, probably due to the lack of reactivity with more sterically demanding secondary and tertiary alkylating reagents (Fig. 1a)[60].

Different from palladium or nickel catalyzed alkylations, the decomposition of alkyl-cobalt intermediates via β-hydrogen elimination is really not a limitation[34]. Whereas cobalt catalysis has demonstrated especially efficient for coupling of alkyl halides, we conceived the unique catalytic characteristics of cobalt could enable the facile constuction of C(sp$^3$)-CF$_2$ bonds. Herein, we report a cobalt-catalyzed difluoroalkylation of tertiary aryl ketones with fluoroalkyl bromides (Fig. 1b). This approach has demonstrated high-catalytic reactivity and broad substrate scope, thus enabling the late-stage fluoroalkylation of biologically active molecules. This strategy offers a solution for facile synthesis of quaternary alkyl difluorides, and provides an efficient way for the synthesis of fluoroalkylated bioactive molecules for drug discovery.

## Results

**Optimization of the co-catalyzed cross-coupling.** Our study commenced with 2-phenyl-3,4-dihydronaphthalen-1(2H)-one **1a** as the pilot substrate and bromodifluoroacetate **2a** as the coupling partner in the presence of catalytic amount of CoBr$_2$ (10 mol%) and dppBz (10 mol%) in THF at −10 °C. Not surprisingly, only trace amount of the desired product **3a** was observed when

**a** Previous work: Ni-catalyzed difluoropropargylation of 1° alkylzinc reagents.

**b** This work: Co-catalyzed difluoroalkylation of tertiary aryl ketones.

**Fig. 1** Transition-metal-catalyzed cross-coupling for facile synthesis of difluoroalkylated alkanes. **a** Ni-catalyzed difluoropropargylation of 1° alkylzinc reagents. **b** Co-catalyzed difluoroalkylation of tertiary aryl ketones

$^t$BuOK was used as the base (Table 1, entry 1). To our delight, 30% yield of **3a** was isolated when zinc metal (0.5 equiv) was added as the key reductant to generate the active cobalt catalyst (Table 1, entry 2)[63]. Further investigation indicated that manganese was not the suitable reductant (Table 1, entry 3), and reducing the amount of zinc resulted in a decline in yield (17%, Table 1, entry 4). To further improve the yield, a careful examination of base was carried out when 0.5 equiv of zinc was added. Whereas inorganic bases gave almost none of the product **3a**, organic bases, including KHMDS and LiHMDS showed good reactivity, and LDA proved as the optimal choice with 90% yield (Table 1, entries 5–7). Different kinds of nitrogen, phosphine, and carbene ligands were next studied, but all gave obviously lower yields or even none of **3a** (Supplementary Table 3). Additionally, careful screening of solvents and cobalt sources indicated THF and CoBr$_2$ were still the best choices (Table 1, entries 8–12 and Supplementary Tables 2 and 6). The reaction temperature was also investigated, which demonstrated that lower temperature even to −30 °C still gave the same excellent yield, but increasing higher temperature (0 °C) resulted in a significant reduction in yield (Table 1, entries 13–14). Finally, control experiments confirmed that almost none of the desired product **3a** was detected in the absence of cobalt catalyst (Table 1, entry 15).

**Scope of the co-catalyzed cross-coupling.** With these optimized condition in hand, various aryl ketone **1** were next tested in this catalytic fluoroalkylation system. As shown in Fig. 2, a variety of cyclic and acyclic aryl ketones were difluoroalkylated successfully, furnishing the desired products **3** with difluoroalkylated quaternary carbon centers in good to excellent yields. To our satisfaction, the examination of α-substituents (R$^1$) of cyclic aryl ketones showed that not only a great number of substituted aryl groups, but also various alkyl groups like Me (**3b**), *n*-Bu (**3c**), Bn (**3d**), and *i*-Pr (**3e**), were smoothly difluoroalkylated with good yields. It should be noted that five- or six-membered carbon rings and even O-containing chromanone were well tolerated in this reaction (**3a-3r**). Indeed, cyclic and acyclic aryl ketones installed with both electron-donating groups, including Me (**3f**, **3q**, **3t**, **3u**), MeO (**3k-3m**, **3v-3×**, **3af**, **3ah**), Me$_2$N (**3ag**), and electron-withdrawing groups, such as F (**3i**, **3ad**, **3af**), Cl (**3g**, **3j**, **3r**, **3ae**), and Br (**3h**, **3n**), on the phenyl rings were fluoroalkylated effectively to give the desired products with acceptable yields in our catalytic system. Remarkably, both cyclic and acyclic aryl ketones containing F (**3i**, **3ad**, **3af**), Cl (**3g**, **3j**, **3r**, **3ae**), Br (**3h**, **3n**) were also suitable coupling partners, which clearly demonstrated good functional group tolerance of our method and offered the synthetic potential for further elaboration by transition-metal-catalyzed coupling reactions. Gratifyingly, acyclic aryl ketones bearing two alkyl groups at the α-position (**3aj**) were well tolerated in our reaction. In addition, aryl ketones containing primary

**Table 1 Cobalt-catalyzed difluoroalkylation: optimization of conditions[a]**

| Entry | Co source | Base | Addi./equiv | Yield (%)[b] |
|---|---|---|---|---|
| 1 | CoBr$_2$ | $^t$BuOK | – | trace |
| 2 | CoBr$_2$ | $^t$BuOK | Zn (0.5) | 30 |
| 3 | CoBr$_2$ | $^t$BuOK | Mn (0.5) | trace |
| 4 | CoBr$_2$ | $^t$BuOK | Zn (0.3) | 17 |
| 5 | CoBr$_2$ | KHMDS | Zn (0.5) | 73 |
| 6 | CoBr$_2$ | LiHMDS | Zn (0.5) | 77 |
| 7 | CoBr$_2$ | LDA | Zn (0.5) | 90 |
| 8 | CoI$_2$ | LDA | Zn (0.5) | 83 |
| 9 | CoCl$_2$•dppe | LDA | Zn (0.5) | 56 |
| 10 | Co(OAc)$_2$•4H$_2$O | LDA | Zn (0.5) | 30 |
| 11 | Co(acac)$_2$ | LDA | Zn (0.5) | 42 |
| 12[c] | CoBr$_2$ | LDA | Zn (0.5) | 50 |
| 13[d] | CoBr$_2$ | LDA | Zn (0.5) | 90 |
| 14[e] | CoBr$_2$ | LDA | Zn (0.5) | 53 |
| 15 | – | LDA | Zn (0.5) | trace |

dppBz = 1,2-bis(diphenylphosphino)benzene
[a]Reaction conditions: **1a** (0.2 mmol, 1.0 equiv), **2a** (3.0 equiv), [Co] (10 mol%), dppBz (10 mol%), base (105 mol%), Zn (0.5 equiv), solvent (2.0 mL), −10 °C, 12 h, N$_2$
[b]Yields of the isolated products given
[c]Dioxane was used as solvent
[d]$T$ = −30 °C
[e]$T$ = 0 °C

and secondary α-C-H bonds had also been investigated in this catalytic system. While no reaction was observed with PhCOCH$_3$, PhCOCH$_2$R (R=Me, Ph) afforded tetrasubstituted mono-fluoroalkenes **8** and **9** in 31% and 41% yield, respectively. In contrast, cyclic aryl ketone furnished a 3,3-difluorofuran-2-one derivative **10** accordingly (Supplementary Figure 186).

To further demonstrate the scope of fluoroalkylating reagents, various kinds of difluoroalkyl bromides were examined in this catalytic system. Not surprisingly, as the analogues with similar reactivity to difluoroacetate **2a**, bromodifluoroacetamides were also compatible with this reaction. Different acetamides, including acyclic diethyl amine, cyclic piperidine, morpholine, and piperazine, could be well tolerated in this transformation with acceptable yields. Additionally, the difluoromethylated arene (**3al**) and heteroarene (**3ak**) could be smoothly coupled to the aryl ketones using this method. Furthermore, the scope of difluoroalkylating reagents had been demonstrated via a diversity of cross-coupling reactions between different kinds of cyclic or acyclic aryl ketones and difluoroalkyl bromides. It is known that the *gem*-difluoromethyl group (CF$_2$) serves widely as a key motif to improve the biological activity of target drug molecules[53,54], and this

method thus demonstrates its application prospect to access diverse difluoroalkylated aryl ketones for drug screen.

Owing to the mild conditions and good functional group tolerance demonstrated in this catalytic system, the synthetic potential of this method was next elucidated via late-stage modification of biologically active molecules with structural diversity. Indeed, donepezil, a well-known acetylcholinesterase inhibitor used to treat the Alzheimer's disease[61], could be difluoroalkylated smoothly with good yields. As shown in Fig. 3, different functional groups, such as ester (**5a**), benzo[d]oxazole (**5b**), and arene (**5c**), in the fluoroalkylating reagents were all well compatible with this catalytic transformation. As an efficient and expedient tactic for the construction of fluorine-containing analogues, these late-stage fluoroalkylations of complex molecules could offer a useful strategy to modify the lead compounds in drug development.

**Mechanistic studies**. To gain some insights into this cobalt-catalyzed difluoroalkylation, a series of control experiments were next performed. Firstly, the subjection of β-piene into the reaction could afford the cycle-opening product **7** in 12% yield along with 35% yield of the desired difluoroalkylated product **3s** (Fig. 4a).

**Fig. 2** Cobalt-catalyzed difluoroalkylation of aryl ketones. **a** Scope of cyclic ketones. **b** Scope of acyclic ketones. **c** Scope of difluoroalkylating reagents. Conditions: **1** (0.2 mmol, 1.0 equiv), **2** (3.0 equiv), CoBr₂ (10 mol%), dppBz (10 mol%), LDA (105 mol%), Zn (0.5 equiv), THF (2.0 mL), −10 °C, 12 h, under N₂ atmosphere. [a] Conditions: 20 mol% of CoBr₂ and 20 mol% of dppBz were used. dppBz = 1,2-bis(diphenylphosphino)benzene

This result indicated that a difluoroalkyl radical was in situ generated in the catalytic cycle. Considering the key role of zinc metal as a reductant for generation of active cobalt species, to understand the catalytically active species in this transformation, Co(I)Cl(PPh₃)₃ had been synthesized and subjected into the reaction[63]. It was found that the addition of zinc metal or not into the standard catalytic conditions had no effect to this transformation (Fig. 4b), giving the desired product **3s** in almost the same

yield (70% and 71%). Whereas zinc metal showed indeed not a strong enough reductant to access Co(0) species[62], these results clearly revealed that Co(I) was the productive catalyst. To further investigate the evidence of generation of the difluoroalkyl radical, β-piene **6** was tested as a radical clock with different cobalt catalysts. As shown in Fig. 4c, the subjection of 1 equiv of Co(II)Br₂/dppBz and zinc were added together into the reaction affording the radical-scavenging product **7**, whereas the omission of zinc

**Fig. 3** Late-stage difluoroalkylation of biologically active molecules. Conditions: **4** (0.2 mmol, 1.0 equiv), **2** (3.0 equiv), CoBr$_2$ (10 mol%), dppBz (10 mol%), LDA (105 mol%), Zn (0.5 equiv), THF (2.0 mL), −10 °C, 12 h, under N$_2$ atmosphere. dppBz = 1,2-bis(diphenylphosphino)benzene

**Fig. 4** Mechanistic studies. **a** Radical trapping experiment with β-piene. **b** Control experiments using Cobalt(I) complex. **c** Studies on generation of the difluoroalkyl radical. **d** Determination of the sequence of transmetallation and activation of fluoroalkyl bromide

furnished none of **7**. Moreover, 1 equiv of premade Co(I)Cl (PPh$_3$)$_3$ could give almost the same result as the combination of Co(II) and zinc used in the reaction system (Fig. 4c). All these results implied that the difluoroalkyl radical was generated by single-electron oxidation of Co(I) with difluoroalkyl bromide **2a**, and zinc served as an efficient reductant to reduce Co(II) to active Co(I) species[62]. The *in-situ* reductive process mentioned not only existed in our system, but also has been reported before[63]. Furthermore, sequential addition of a mixture of **1a** (1 equiv)/LDA (1 equiv), and then fluoroalkylating reagent **2a** (3 equiv) gave a comparable yield in 72%, but the reverse order of addition with the same reagents gave only 10% yield of the desired product **3s** (Fig. 4d). These results indicated the transmetallation step should occur before the activation of fluoroalkyl bromide.

Based on all of these results and the previous reports[64–66], a proposed mechanism of Co(I) initiated cross-coupling was described in Fig. 5. First of all, reduction of Co(II) by zinc metal afforded the catalytically active Co(I) species **A** to start the cycle[66]. Transmetallation between Co(I) **A** and enol anion **B**,

**Fig. 5** Proposed mechanism. The possible reaction pathway based on our studies and the previous literatures

which was in situ generated with the assistance of LDA, afforded the corresponding Co(I) complex **C** and **D**. A single electron oxidation of Co(I) **C** by fluoroalkylating reagent **2** generated the corresponding radical and Co(II) species, which was further transformed to Co(III) intermediate **F** after the following radical oxidation. At last, the reductive elimination of **F** furnished the final product **3** and regenerated Co(I) species to enter the next catalytic circle.

## Discussion

In summary, we have developed a difluoroalkylation of tertiary C–H bonds through cobalt-catalyzed cross-coupling between aryl ketones and fluoroalkyl bromides. Mechanistic investigations indicated this C–H fluoroalkylation proceeds via a Co(I)/Co(III) catalytic cycle involving an in situ generated difluoroalkyl radical. This method has demonstrated mild conditions, broad substrate scope, and thus enabled the late-stage difluoroalkylation of complex molecules. This strategy will offer a solution for facile synthesis of quaternary alkyl difluorides. Further application of this method to fluorine-containing modification of complex biologically active molecules is still underway in our laboratory.

## Methods

**General procedure for the cobalt-catalyzed cross-coupling**. To a 50 mL of Schlenk tube was added aryl ketone **1** (1.0 equiv, 0.2 mmol), CoBr$_2$ (10 mol %, 0.02 mmol) and dppBz (10 mol %, 0.02 mmol) under air, followed by Zn (0.5 equiv, 0.1 mmol). The mixture was evacuated and backfilled with N$_2$ (three times). THF (2 mL) was added then followed by LDA (105 mol%, 0.21 mmol) subsequently. The Schlenk tube was then sealed with a Teflon lined cap and put into a cooled bath (−10 °C). After stirring for 5 min, bormdifluoroacetate **2a** (3.0 equiv, 0.6 mmol) was added to the reaction mixture, and the Schlenk tube was then resealed with a Teflon lined cap and put back into the cooled bath (−10 °C). After stirring for another 12 h, the reaction mixture was diluted with ethyl acetate (5 mL). The solvent was removed under reduced pressure, and the residue was purified by flash column chromatography on silica gel to give the desired product.

## Data availability

The authors declare that all the data supporting the findings of this research are available within the article and its supplementary information.

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

## Acknowledgements

We gratefully acknowledge the Strategic Priority Research Program of the Chinese Academy of Sciences (Grant No. XDB20000000), the National Basic Research Program of China (973 Program 2015CB856600), the National Science Foundation of China (21772187, 21522208) for financial support.

## Author contributions

C.L. and Y.-X.C. designed and performed the experiments. R.W., Y.-N.W., and Q.L. helped to complete the experiments. X.-S.W. directed the project and wrote the manuscript. All authors interpreted the results on the manuscript.

## Additional information

**Competing interests:** The authors declare no competing interests.

