## [Peer Review File · Nature Communications]

Reviewer #1 (Remarks to the Author):

This manuscript discloses the conventional Co-catalyzed difluoroalkylation and I consider that it is worthwhile publishing in Nature Communications.

My comments are

(i) The authors showed the reactions at higher temperature afforded lower yields. What happened at the high temperature? Mass balance is OK?

(ii) In Figure 3, the authors proposed mechanism. I cannot understand the charge balance in the process of C + Br-Rf to E and E to F. I strongly recommend the authors indicate the charge of E and F.

(iii) The next paper seems important in references.

Ohtsuka, Y.; Yamakawa, T.

Direct Ethoxycarbonyldifluoromethylation of Aromatic Compounds Using Fenton Reagent

Tetrahedron, 2011, 67 (12), 2323 – 2331.

(iv) I found the careless mistakes as follows:

In Figure 1: "Co-Catalyzed" should be "Co-catalyzed"

In page 7 4 lines from bottom: "znic" should be "zinc"

In Table 1: the column of Additives is misaligned.

I recommend the authors check again carefully in the whole manuscript.

Reviewer #2 (Remarks to the Author):

The novelty of the work consists of the combination of the generally known cross-coupling of aryl ketones with a fluoroalkyl bromides under a system modified from predecessors' conditions. The reaction features a certain scope with low to good yields. The manuscript is generally written in good English and the Supporting Information seems to confirm the proposed structures as well as the sufficient purity of the compounds isolated.

As we know, fluoroalkylation reaction is a direct strategy for effective synthesis of fluoroalkylated molecule, but in this manuscript both general reactivities have been a substantial amount of reported previously. On the other hand, the similar products had also been reported by Ando (JFC 2010, 131, 86) and Wang (OL 2017, 19, 5653) utilizing other mild reaction conditions. Moreover, the scope of these transformations is limited to easily formed enol compounds, other containing tertiary C-H bonds of molecules without ketone should be tested. As there have been several reports of cross-coupling of aryl ketones of this type, it is difficult to see where this work fits in with regard to novelty. What's more, the proposed mechanism is confusing and incomplete. How does the free radicals generate? Any evidences? Since the above mentioned reasons, I do not support the present technology for publication in Nature Communications.

Reviewer #3 (Remarks to the Author):

Recently, the preparation of difluoroalkylated compounds has received increasing attention and important progress has been achieved. However, most of the developed methods have been focused on the construction of Ar-CF₂R bonds. To date, the construction of Alkyl-CF₂R bond via transition-metal-catalyzed cross-coupling remains underdeveloped and only rare examples have been reported (ref 72). In this regards, the manuscript by Wang and coworkers describes a cobalt-catalyzed cross-coupling between aryl ketones and difluoroalkyl bromides. The reaction proceeds at -10 °C with LDA as a base to generate lithium enolates for the cross-coupling. Different active difluoroalkyl bromides, including bromodifluoroacetate, bromodifluoroacetamides and (hetero)aryldifluoromethyl bromides, are applicable to the reaction, providing the corresponding products with moderate to high yields. To demonstrate the utility of the protocol, reactions of bioactive molecule 4 with difluoroalkyl bromides were also conducted, which may have potential applications in medicinal chemistry. Mechanistic studies reveal that the present cobalt-catalyzed process may be initiated by Co(I), and a Co(I/III) is involved in the reaction. Since this is the first example of cobalt-catalyzed difluoroalkylation of aryl ketones and it provides the opportunity for the enantioselective difluoroalkylation catalyzed by transition-metal, the manuscript can be published in Nat. Commun, but the authors should address the following issues.

1) The authors emphasize that the requirement of additional step to synthesize alkyl zinc reagents is the drawback of the nickel-catalyzed difluoroalkylation reported by Zhang, but their method have the advantage of step-economy, namely, direct C-H bond difluoroalkylation of aryl ketones. However, a LDA is required to generate lithium enolate for the cross-coupling, which is also an additional step, thus it is not step-economy. Additionally, it is well-known that the cleavage of α C-H bonds of the ketones are very easy, the authors should not emphasize the direct C-H difluoroalkylation of ketones is the advantage of the reaction. Thus, the wording should be revised. Furthermore, the authors should be aware that some base and nucleophile sensitive functional

groups, such as aldehyde, ketone, CN, and so on, are not compatible with their reaction conditions because of the use of LDA.

2) The authors emphasize it is a challenge for the synthesis of difluoroalkyl substituted ketones bearing a quaternary carbon center. However, the referee concerns that how about aryl-ketone bearing no branched substituents, such as ArCOCH₃, ArCOCH₂CH₃ and aryl cyclic ketones. The authors should examine these substrates and provide the results in the text.

3) Only aryl ketones were applicable to the reaction. How about alkyl-alkyl ketones, AlkylCOAlkyl?

4) For the mechanistic studies, it would be better to use MS to analyze the intermediate generated between Co(I) and lithium enolate.

5) The first example of nickel-catalyzed difluoroalkylation of aromatics via cross-coupling should be cited (*Angew. Chem. Int. Ed.* 2014, 53, 9909, *Nat. Commn.* 2018, 9, 1170). Additionally, the review of transition-metal-catalyzed difluoroalkylation via cross-coupling by difluoroalkyl halides should be cited (*Acc. Chem. Res.* 2018, DOI: 10.1021/acs.accounts.8b00230).

6) The catalytic enantioselective difluoroalkylations of aldehydes and ketones with difluoroalkyl halides have been reported, the authors should cite these papers and mentioned in the text (*Tetrahedron Asymmetry*, 2008, 19, 2633; *JACS*, 2009, 131, 10875).

7) "gem-difluoromethyl groups" should be "difluoromethylene group"

8) Table 3. "Late-stage difluoromethylation" should be "Late-stage difluoroalkylation"

Reviewer 1:

1. The authors showed the reactions at higher temperature afforded lower yields. What happened at the high temperature? Mass balance is OK?

We have performed the reaction at 50 °C, as suggested. The desired difluoroalkylated product **3s** was afforded in 24% yield along with 68% recovery of raw material **1s**, and no by-products could be isolated from the reaction system.

2. In Figure 3, the authors proposed mechanism. I cannot understand the charge balance in the process of C + Br-Rf to E and E to F. I strongly recommend the authors indicate the charge of E and F.

The charge balance mentioned has been corrected, and the charge of E and F was indicated as II and III, respectively, as shown in Figure 2.

3. The next paper seems important in references. Ohtsuka, Y.; Yamakawa, T. Direct Ethoxycarbonyldifluoromethylation of Aromatic Compounds Using Fenton Reagent, *Tetrahedron*, 2011, 67 (12), 2323 – 2331.

This reference has been added into the manuscript as ref. 66, as suggested.

4. I found the careless mistakes as follows:

In Figure 1: “Co-Catalyzed” should be “Co-catalyzed”

In page 7 4 lines from bottom: “znic” should be “zinc”

In Table 1: the column of Additives is misaligned.

All of the mentioned mistakes have been corrected, as suggested.

Reviewer 2:

1. As we know, fluoroalkylation reaction is a direct strategy for effective synthesis of fluoroalkylated molecule, but in this manuscript both general reactivities have been a substantial amount of reported previously. On the other hand, the similar products had also been reported by Ando (*JFC* 2010, 131, 86) and Wang (*OL* 2017, 19, 5653) utilizing other mild reaction conditions.

Actually, the mentioned reports from Ando and Wang used premade enamine or

silyl enol ether as the substrate, and our work used ketones directly for difluoroalkylation. It should be noted that our manuscript described the first example of cobalt-catalysis for fluoroalkylation of alkyl substrates.

- Moreover, the scope of these transformations is limited to easily formed enol compounds, other containing tertiary C-H bonds of molecules without ketone should be tested.

As suggested, 2,2-diphenylacetonitrile and ethyl 2,2-diphenylacetate, both containing tertiary C-H bond, have been tested with the standard conditions. To our surprise, both substrates furnished only quaternary alkyl monofluorides in pretty low yields. We reckoned that a defluorination reaction happened after the difluoroalkylation to afford the final alkyl monofluorides. While the mechanism of this transformation is still unknown and the products are not the expected difluorides as in this manuscript, the relevant research works are still underway in our lab.

- What's more, the proposed mechanism is confusing and incomplete. How does the free radicals generate? Any evidences?

As shown in Eq. 1, Scheme 2, the subjection of β -piene **6**, a known radical clock, into the standard conditions could furnish the cycle-opening product **7** in 12% yield, which indicated that a difluoroalkyl radical was involved in the catalytic cycle. To get some evidence that how the difluoroalkyl radical generated, β -piene **6** was further tested as a radical capture with different cobalt catalysts. Actually, the addition of 1 equiv of Co(II) and zinc into the reaction could afford the radical-scavenging product **7**, while the omission of zinc furnished none of **7**. Additionally, the use of 1 equiv of Co(I)Cl(PPh₃)₃ could give almost the same

result as the combination of Co(II) and zinc. All of these results showed that the difluoroalkyl radical was generated by single-electron-oxidation of Co(I) with difluoroalkyl bromide **2a**, and zinc served as an efficient reductant to reduce Co(II) to active Co(I) species.

These results have been added into the manuscript, and described as, “To further get some evidences that how the difluoroalkyl radical generated, β -piene **6** was tested as a radical clock with different cobalt catalysts. As shown in Eq. 3, the subjection of 1 equiv of Co(II)Br₂/dppBz and zinc were added into the reaction together and afforded the radical-scavenging product **7**, while the omission of zinc furnished none of **7**. Moreover, 1 equiv of premade Co(I)Cl(PPh₃)₃ could give almost the same result as the combination of Co(II) and zinc was used in the reaction system (Eq. 3). All these results implied that the difluoroalkyl radical was generated by single-electron-oxidation of Co(I) with difluoroalkyl bromide **2a**, and zinc served as an efficient reductant to reduce Co(II) to active Co(I) species.”

Reviewer 3:

1. The authors emphasize that the requirement of additional step to synthesize alkyl zinc reagents is the drawback of the nickel-catalyzed difluoroalkylation reported by Zhang, but their method have the advantage of step-economy, namely, direct C-H bond difluoroalkylation of aryl ketones. However, a LDA is required to generate lithium enolate for the cross-coupling, which is also an additional step, thus it is not step-economy. Additionally, it is well-known that the cleavage of C-H bonds of the ketones are very easy, the authors should not emphasize the direct C-H difluoroalkylation of ketones is the advantage of the reaction. Thus, the wording should be revised. Furthermore, the authors should be aware that some base and nucleophile sensitive functional groups, such as aldehyde, ketone, CN, and so on, are not compatible with their reaction conditions because of the use of LDA.

Thanks for these suggestions. We have deleted the corresponding comments from the manuscript to avoid controversy, including the emphasis of both step-economy and direct C-H difluoroalkylation. Meanwhile, the title of this manuscript has also been revised to “Cobalt-Catalyzed Difluoroalkylation of Tertiary α -C-H Bonds of Aryl Ketones for Facile Synthesis of Quaternary Alkyl Difluorides”. Actually, “ α -C-H bond of ketones” is a widely known and used term for this kind of activated C-H bonds.

2. The authors emphasize it is a challenge for the synthesis of difluoroalkyl substituted ketones bearing a quaternary carbon center. However, the referee concerns that how about aryl-ketone bearing no branched substituents, such as ArCOCH₃, ArCOCH₂CH₃ and aryl cyclic ketones. The authors should examine these substrates and provide the results in the text.

As suggested, aryl-ketones bearing no branched substituents, including PhCOCH₃, PhCOCH₂R and aryl cyclic ketones, have been investigated under the standard conditions. While no reaction was observed with PhCOCH₃, PhCOCH₂R (R = Me, Ph) containing secondary α -C-H bond afforded tetrasubstituted monofluoroalkenes in 31-41% yields. In contrast, cyclic aryl ketone furnished the corresponding 3,3-difluorofuran-2-one derivative **10**. These tetrasubstituted monofluoroalkenes and 3,3-difluorofuran-2-one derivative were speculated to be generated by defluorination or alcoholysis of the enolate anion stemming from the normal difluoroalkylated product.

These results have been added into the Supporting Information, and described in the manuscript as, “In addition, aryl ketones containing primary and secondary α -C-H bonds have also been investigated in this catalytic system. While no reaction was observed for PhCOCH₃, PhCOCH₂R (R = Me, Ph) afforded tetrasubstituted monofluoroalkenes **8** and **9** in 31% and 41% yield, respectively. In contrast, cyclic aryl ketone furnished a 3,3-difluorofuran-2-one derivative **10**, accordingly (For details, see the supporting information).”

3. Only aryl ketones were applicable to the reaction. How about alkyl-alkyl ketones, AlkylCOAlkyl?

As suggested, alkyl-alkyl ketone has been investigated under the standard

conditions of our catalytic system, but gave none of the desired fluoroalkylated product.

4. For the mechanistic studies, it would be better to use MS to analyze the intermediate generated between Co(I) and lithium enolate.

Thanks for this nice suggestion. To determine the intermediate generated from the transmetallation of Co(I) with lithium enolate, we have conducted high resolution mass spectra analysis of the in-situ-generated cobalt(I) complex C for several times. Unfortunately, the key species has not been detected, possibly due to its extreme instability.

5. The first example of nickel-catalyzed difluoroalkylation of aromatics via cross-coupling should be cited (Angew. Chem. Int. Ed. 2014, 53, 9909, Nat. Commn. 2018, 9, 1170). Additionally, the review of transition-metal-catalyzed difluoroalkylation via cross-coupling by difluoroalkyl halides should be cited (Acc. Chem. Res. 2018, DOI: 10.1021/acs.accounts.8b00230). The catalytic enantioselective difluoroalkylations of aldehydes and ketones with difluoroalkyl halides have been reported, the authors should cite these papers and mentioned in the text (Tetrahedron Asymmetry, 2008, 19, 2633; JACS, 2009, 131, 10875).

All these mentioned papers have been added into the manuscript as ref. 9, 10, 11, 61 and 62, as suggested.

6. “gem-difluoromethyl groups” should be “difluoromethylene group” and Table 3. “Late-stage difluoromethylation” should be “Late-stage difluoroalkylation”

The corresponding mistakes have been revised, as suggested.

Reviewer #1 (Remarks to the Author):

I consider that the manuscript has been well revised according to the referee's comments.

I evaluate this manuscript should be accepted without revision.

Reviewer #2 (Remarks to the Author):

Based on the authors' original manuscript on the "Cobalt-Catalyzed Difluoroalkylation of Tertiary C-H Bonds of Aryl Ketones" Wang and co-workers have submitted a revised manuscript and have responded to the referees comments. While some suggested revisions have been taken into account, the work still features several shortcomings:

The reaction is similar to SN2 reaction, and there are a substantial amount of reports on ketones and halohydrocarbon. Actually, bromodifluoroacetate as electrophilic reagents had been used in some substrate synthesis, such as phenol/phenthinol (Org. Lett. 2016, 18, 4570; Angew. Chem. 2015, 127, 10129 See SI) as nucleophile. In addition, the alkylation of ketones had been reported (J. Am. Chem. Soc. 2016, 138, 9935; Angew. Chem. Int. Ed. 2017, 56, 15035; See SI and so on). What's more, the authors consider it is a challenge for the synthesis of difluoroalkyl substituted ketones bearing a quaternary carbon center, but asymmetric perfluoroalkylation of β -ketoesters had been developed previously (J. Am. Chem. Soc. 2015, 137, 5678; Org. Lett. 2018, 20, 461).

The scope of these transformations is limited to tertiary aryl ketones. Alkyl-alkyl ketones and aryl-ketone bearing no branched substituents give unsatisfactory results.

In consideration of the reasons above, this reviewer to recommend the publication of this manuscript in other journals rather than the Nature Communications after revision.

Reviewer #3 (Remarks to the Author):

The authors have addressed all the issues raised by the referees. The current revised version can be published in Nat. Commun.

Reviewer 1:

1. The authors showed the reactions at higher temperature afforded lower yields. What happened at the high temperature? Mass balance is OK?

Response: We have performed the reaction at 50 °C, as suggested. The desired difluoroalkylated product **3s** was afforded in 24% yield along with 68% recovery of raw material **1s**, and no by-products could be isolated from the reaction system.

2. In Figure 3, the authors proposed mechanism. I cannot understand the charge balance in the process of C + Br-R_f to E and E to F. I strongly recommend the authors indicate the charge of E and F.

Response: The charge balance mentioned has been corrected, and the charge of E and F was indicated as II and III, respectively, as shown in Figure 5.

3. The next paper seems important in references. Ohtsuka, Y.; Yamakawa, T. Direct Ethoxycarbonyldifluoromethylation of Aromatic Compounds Using Fenton Reagent, *Tetrahedron*, 2011, 67 (12), 2323 – 2331.

Response: This reference has been added into the manuscript as ref. 48, as suggested.

4. I found the careless mistakes as follows:

In Figure 1: “Co-Catalyzed” should be “Co-catalyzed”

In page 7 4 lines from bottom: “znic” should be “zinc”

In Table 1: the column of Additives is misaligned.

Response: All of the mentioned mistakes have been corrected, as suggested.

Reviewer 2:

1. As we know, fluoroalkylation reaction is a direct strategy for effective synthesis of fluoroalkylated molecule, but in this manuscript both general reactivities have been a substantial amount of reported previously. On the other hand, the similar products had also been reported by Ando (*JFC* 2010, 131, 86) and Wang (*OL* 2017, 19, 5653) utilizing other mild reaction conditions.

Response: Actually, the mentioned reports from Ando and Wang used pre-made enamine or silyl enol ether as the substrate, and our work used ketones directly for difluoroalkylation. It should be noted that our manuscript described the first example of cobalt-catalysis for fluoroalkylation of alkyl substrates.

2. Moreover, the scope of these transformations is limited to easily formed enol compounds, other containing tertiary C-H bonds of molecules without ketone should be tested.

Response: As suggested, 2,2-diphenylacetonitrile and ethyl 2,2-diphenylacetate, both containing tertiary C-H bond, have been tested with the standard conditions. To our surprise, both substrates furnished only quaternary alkyl monofluorides in pretty low yields. We reckoned that a defluorination reaction happened after the difluoroalkylation to afford the final alkyl monofluorides. While the mechanism of this transformation is still unknown and the products are not the expected difluorides as in this manuscript, the relevant research works are still underway in our lab.

3. What's more, the proposed mechanism is confusing and incomplete. How does the free radicals generate? Any evidences?

Response: As shown in Eq. c, Figure 4, the subjection of β -piene **6**, a known radical clock, into the standard conditions could furnish the cycle-opening product **7** in 12% yield, which indicated that a difluoroalkyl radical was involved in the catalytic cycle. To get some evidence that how the difluoroalkyl radical generated, β -piene **6** was further tested as a radical capture with different cobalt catalysts. Actually, the addition of 1 equiv of Co(II) and zinc into the reaction could afford the radical-scavenging product **7**, while the omission of zinc furnished none of **7**.

Additionally, the use of 1 equiv of $\text{Co(I)Cl(PPh}_3)_3$ could give almost the same result as the combination of Co(II) and zinc. All of these results showed that the difluoroalkyl radical was generated by single-electron-oxidation of Co(I) with difluoroalkyl bromide **2a**, and zinc served as an efficient reductant to reduce Co(II) to active Co(I) species.

These results have been added into the manuscript, and described as, “To further get some evidences that how the difluoroalkyl radical generated, β -piene **6** was tested as a radical clock with different cobalt catalysts. As shown in Eq. c, Figure 4, the subsection of 1 equiv of $\text{Co(II)Br}_2/\text{dppBz}$ and zinc were added into the reaction together and afforded the radical-scavenging product **7**, while the omission of zinc furnished none of **7**. Moreover, 1 equiv of premade $\text{Co(I)Cl(PPh}_3)_3$ could give almost the same result as the combination of Co(II) and zinc was used in the reaction system (Figure 4, C). All these results implied that the difluoroalkyl radical was generated by single-electron-oxidation of Co(I) with difluoroalkyl bromide **2a**, and zinc served as an efficient reductant to reduce Co(II) to active Co(I) species.”

Reviewer 3:

1. The authors emphasize that the requirement of additional step to synthesize alkyl zinc reagents is the drawback of the nickel-catalyzed difluoroalkylation reported by Zhang, but their method have the advantage of step-economy, namely, direct C-H bond difluoroalkylation of aryl ketones. However, a LDA is required to generate lithium enolate for the cross-coupling, which is also an additional step, thus it is not step-economy. Additionally, it is well-known that the cleavage of C-H bonds of the ketones are very easy, the authors should not emphasize the direct C-H difluoroalkylation of ketones is the advantage of the reaction. Thus, the wording should be revised. Furthermore, the authors should be aware that some base and nucleophile sensitive functional groups, such as aldehyde, ketone, CN, and so on, are not compatible with their reaction conditions because of the use of LDA.

Response: Thanks for these suggestions. We have deleted the corresponding comments from the manuscript to avoid controversy, including the emphasis of both step-economy and direct C-H difluoroalkylation. Meanwhile, the title of this manuscript has also been revised to “Cobalt-Catalyzed Difluoroalkylation of Tertiary Aryl Ketones for Facile Synthesis of Quaternary Alkyl Difluorides”.

Actually, “Tertiary Aryl Ketones” is a widely known and used term for this kind of activated C–H bonds.

2. The authors emphasize it is a challenge for the synthesis of difluoroalkyl substituted ketones bearing a quaternary carbon center. However, the referee concerns that how about aryl-ketone bearing no branched substituents, such as ArCOCH_3 , $\text{ArCOCH}_2\text{CH}_3$ and aryl cyclic ketones. The authors should examine these substrates and provide the results in the text.

Response: As suggested, aryl-ketones bearing no branched substituents, including PhCOCH_3 , PhCOCH_2R and aryl cyclic ketones, have been investigated under the standard conditions. While no reaction was observed with PhCOCH_3 , PhCOCH_2R ($\text{R} = \text{Me}, \text{Ph}$) containing secondary $\alpha\text{-C-H}$ bond afforded tetrasubstituted monofluoroalkenes in 31-41% yields. In contrast, cyclic aryl ketone furnished the corresponding 3,3-difluorofuran-2-one derivative **10**. These tetrasubstituted monofluoroalkenes and 3,3-difluorofuran-2-one derivative were speculated to be generated by defluorination or alcoholysis of the enolate anion stemming from the normal difluoroalkylated product.

These results have been added into the supplementary information, and described in the manuscript as, “In addition, aryl ketones containing primary and secondary $\alpha\text{-C-H}$ bonds have also been investigated in this catalytic system. While no reaction was observed for PhCOCH_3 , PhCOCH_2R ($\text{R} = \text{Me}, \text{Ph}$) afforded tetrasubstituted monofluoroalkenes **8** and **9** in 31% and 41% yield, respectively. In contrast, cyclic aryl ketone furnished a 3,3-difluorofuran-2-one derivative **10**, accordingly (For details, see the supplementary information).”

3. Only aryl ketones were applicable to the reaction. How about alkyl-alkyl ketones, AlkylCOAlkyl ?

Response: As suggested, alkyl-alkyl ketone has been investigated under the standard conditions of our catalytic system, but gave none of the desired fluoroalkylated product.

- For the mechanistic studies, it would be better to use MS to analyze the intermediate generated between Co(I) and lithium enolate.

Response: Thanks for this nice suggestion. To determine the intermediate generated from the transmetalation of Co(I) with lithium enolate, we have conducted high resolution mass spectra analysis of the in-situ-generated cobalt(I) complex C for several times. Unfortunately, the key species has not been detected, possibly due to its extreme instability.

- The first example of nickel-catalyzed difluoroalkylation of aromatics via cross-coupling should be cited (Angew. Chem. Int. Ed. 2014, 53, 9909, Nat. Commn. 2018, 9, 1170). Additionally, the review of transition-metal-catalyzed difluoroalkylation via cross-coupling by difluoroalkyl halides should be cited (Acc. Chem. Res. 2018, DOI: 10.1021/acs.accounts.8b00230). The catalytic enantioselective difluoroalkylations of aldehydes and ketones with difluoroalkyl halides have been reported, the authors should cite these papers and mentioned in the text (Tetrahedron Asymmetry, 2008, 19, 2633; JACS, 2009, 131, 10875).

Response: All these mentioned papers have been added into the manuscript as ref. 43, 44, 9, 11 and 10, as suggested.

- “gem-difluoromethyl groups” should be “difluoromethylene group” and Table 3. “Late-stage difluoromethylation” should be “Late-stage difluoroalkylation”

Response: The corresponding mistakes have been revised, as suggested.

Point-by-point response to the referee's

Reviewer 1:

1. I consider that the manuscript has been well revised according to the referee's comments. I evaluate this manuscript should be accepted without revision.

Response: Thanks for the comments and suggestions during the submission.

Reviewer 2:

1. Based on the authors' original manuscript on the "Cobalt-Catalyzed Difluoroalkylation of Tertiary C-H Bonds of Aryl Ketones" Wang and co-workers have submitted a revised manuscript and have responded to the referees comments. While some suggested revisions have been taken into account, the work still features several shortcomings:

The reaction is similar to S_N2 reaction, and there are a substantial amount of reports on ketones and halohydrocarbon. Actually, bromodifluoroacetate as electrophilic reagents had been used in some substrate synthesis, such as phenol/phenthionol (Org. Lett. 2016, 18, 4570; Angew. Chem. 2015, 127, 10129 See SI) as nucleophile. In addition, the alkylation of ketones had been reported (J. Am. Chem. Soc. 2016, 138, 9935; Angew. Chem. Int. Ed. 2017, 56, 15035; See SI and so on). What's more, the authors consider it is a challenge for the synthesis of difluoroalkyl substituted ketones bearing a quaternary carbon center, but asymmetric perfluoroalkylation of β -ketoesters had been developed previously (J. Am. Chem. Soc. 2015, 137, 5678; Org. Lett. 2018, 20, 461).

The scope of these transformations is limited to tertiary aryl ketones. Alkyl-alkyl ketones and aryl-ketone bearing no branched substituents give unsatisfactory results.

Response: Thanks for the comments and suggestions during the submission.

Compared with the non-fluorinated halohydrocarbon, fluoroalkyl halides showed totally different reactivity in the organic transformations. As shown in this manuscript, the control experiments confirmed that almost none of the desired product **3a** was detected in the absence of cobalt catalyst (entry 24, Table 1), which demonstrated this transformation is not a S_N2 reaction. Additionally, different from our catalytic system, the asymmetric perfluoroalkylation worked only for β -ketoesters via a direct radical capture process, and no aryl ketones have

been demonstrated in both reports. Finally, there is no common catalytic system works for all substrates and the development of novel catalytic system is always on the way in organic synthesis.

Reviewer 3:

1. The authors have addressed all the issues raised by the referees. The current revised version can be published in Nat. Commun.

Response: Thanks for the comments and suggestions during the submission.